

# Exploring the potential of online social listening for noncommunicable disease monitoring

Diana Braga[1], Inês Silva[1], Rafaela Rosário[2], Paulo Novais[3], Hugo Peixoto[3] and José Machado[3]

[1] Department of Informatics, University of Minho, Braga, Portugal
[2] School of Nursing, University of Minho, Braga, Portugal
[3] Algoritmi/LASI, University of Minho, Braga, Portugal

Corresponding authors
Hugo Peixoto,
hpeixoto@di.uminho.pt
José Machado, jmac@di.uminho.pt

## ABSTRACT

Noncommunicable diseases (NCDs) are a significant global health challenge, claiming about 41 million lives annually. Early establishment of healthy habits is vital as childhood behaviors often persist into adulthood, affecting long-term well-being. However, pervasive health misinformation on social media exacerbates the challenge of addressing NCDs. The vast online information exposes individuals to misinformation, leading to uninformed health decisions. Countering this misinformation is crucial to promote accurate understanding and preventive strategies for NCDs, improving public health outcomes. To address this, the study proposes a system using online social listening (OSL) to collect and analyze social media data, focusing on children's nutrition, physical exercise, sleep patterns, and related NCD risk factors. This platform aids healthcare professionals in recognizing and responding to online misinformation, facilitating informed decision-making. Collaboration with parents, teachers, and healthcare providers aims to instill healthy habits in children from an early age. Utilizing the Twitter Application Programming Interface (API), the study collected data on NCDs, their risk factors, and their impact on children. Despite challenges from recent Twitter API policy changes, the methodology remains adaptable. Additionally, the study integrates diverse data sources, including traditional news outlets like PressReader, providing comprehensive coverage of health issues. Analysis comparing data from PressReader and Twitter underscores differences in discussion frequency and nature, emphasizing the need to leverage insights from various sources. The results highlight the effectiveness of the OSL system in identifying prevalent health topics, benefiting healthcare professionals. This collaborative approach positions the system as a valuable tool for addressing NCDs and promoting well-being. The study lays a foundation for future research, suggesting expansions to include additional platforms and languages, as well as advanced features like sentiment analysis.

## INTRODUCTION

Noncommunicable diseases (NCDs) stand as persistent health conditions not transmitted from person to person, unlike infectious diseases. They play a significant role in the top 10 causes of death globally, spanning low-, middle-, and high-income countries, as identified by the World Health Organization (WHO) (*Budreviciute et al., 2020*; *WHO, 2020*). According to WHO statistics on "NCD: Mortality" (*WHO, 2023a*), it is evident that NCDs have a profound global impact, leading to the demise of 41 million individuals annually. Remarkably, over 15 million lives are claimed by NCDs every year within the age range of 30 to 69 years, with an overwhelming 85% of "premature" deaths.

Furthermore, NCDs not only represent a primary cause of mortality but also contribute substantially to the loss of disability-adjusted life years (DALYs) on a global scale, equivalent to the loss of 1 year of full health. Examples of NCDs encompass cardiovascular diseases (coronary artery disease, heart failure, and stroke), cancer, chronic respiratory diseases (asthma and chronic obstructive pulmonary disease), diabetes, and mental health conditions, among others (*WHO, 2022b*).

Of growing concern is the increasing prevalence of NCDs among children, a demographic that has traditionally been less associated with these conditions. Childhood obesity, for instance, is a significant risk factor for the early onset of metabolic syndrome, type 2 diabetes, and cardiovascular diseases (*GBD 2021 US Obesity Forecasting Collaborators, 2024*). Research indicates that the prevalence of obesity and overweight among children has risen dramatically over the past few decades, influenced by sedentary lifestyles, unhealthy diets, and socioeconomic factors (*Lister et al., 2023*). Moreover, childhood exposure to modifiable risk factors, such as poor dietary habits and physical inactivity, contributes to a higher likelihood of these individuals developing chronic conditions in adulthood (*Laddu et al., 2024*).

NCDs are influenced by various factors, including lifestyle, genetics, environment, and sociodemographics. While certain risk factors like age and family history are unmodifiable, many others, such as an unhealthy diet, physical inactivity, tobacco use, harmful alcohol consumption, and sleep deprivation, can be modified through lifestyle changes and interventions. An unhealthy diet, involving the consumption of fried, processed, and sugary meals, elevates the risk of heart disease, diabetes, obesity, hypertension, and specific cancers. Physical inactivity is another modifiable risk factor associated with heart disease, diabetes, and obesity. Tobacco use increases the risk of lung cancer and chronic respiratory diseases, while excessive alcohol use can contribute to liver disease, cancer, and cardiovascular issues (*Budreviciute et al., 2020*).

Addressing modifiable risk factors is crucial, with the potential to prevent 80% of heart diseases, strokes, diabetes, and 40% of cancer cases (*Davagdorj et al., 2021*). While many risk factors are modifiable through public health interventions and early screenings, comprehensive best practices in this realm are still lacking. Given that NCDs often trace their origins to childhood habits with potential lasting effects into adulthood, preventive measures and the promotion of healthy lifestyles in children are crucial for long-term benefits to both public health and healthcare systems (*Yan & Mi, 2021*). Beyond these

challenges, the pervasiveness of misinformation and its amplification through social media platforms adds a new layer of complexity.

The global population of Internet users surpassed 5 billion unique individuals in October 2022, with over 70% utilizing the Internet for healthcare-related information (*Li et al., 2020*; *DataReportal, 2018*). In January 2023, Portugal had 8.73 million Internet users, constituting 85.1% of the total population, with "Researching Health Issues and Healthcare Products" ranking among the top 15 reasons for Internet usage among the Portuguese (*DataReportal, 2023*). Concurrently, social media usage is on the rise globally, especially in contexts related to health issues (*Suarez-Lledo & Alvarez-Galvez, 2021*). With the simplicity of generating online content and the immense volume of information circulated daily, a notable health concern arises. Individuals may easily become misinformed, not only due to the overwhelming amount of health information but also because some of it is false or inaccurate (*Swire-Thompson & Lazer, 2020*).

Children aged 6–10 years usually have limited direct access to smartphones or social media platforms. Although, it is important to note that this group is often exposed to online media through indirect means, such as through shared devices with parents or caregivers. Research shows that the younger demographic is increasingly engaging with digital content, with many children gaining access to the internet and social media, often under parental supervision (*Dinleyici et al., 2016*). A study by the American Academy of Pediatrics found that children aged 6–11 spend an average of 2–3 h per day on digital media, including educational content and entertainment, which may also include exposure to health-related information (*Qi, Yan & Yin, 2023*).

Furthermore, adults such as parents, educators, and healthcare providers, who influence children's behaviours, are highly active on social media platforms like *Business of Apps (2024)*. These groups play a crucial role in shaping the health-related narratives and decisions that affect children, making social media data highly relevant for tracking the broader public discussions about children's health and NCDs.

Health misinformation involves the circulation of false or misleading information about health, shared without intent to cause harm, posing potential serious consequences for individuals and communities (*Swire-Thompson & Lazer, 2020*). It can result in ill-informed health choices, disease transmission, setbacks in public health, economic repercussions, and adverse effects on mental well-being (*WHO, 2022a*). Research often focuses on platforms like Twitter and YouTube, yet health misinformation extends to various platforms and offline channels.

Online social listening (OSL) can play a vital role in identifying information gaps and viral misinformation narratives. It involves the continuous monitoring and analysis of conversations and interactions across social media, forums, and online platforms. This process allows to gather data, understand public sentiment, identify trends, and inform decision-making, particularly in the context of online discourse about specific topics (*Lutkevich & Hildreth, 2022*). With 4.76 billion active global social media users, constituting 59.4% of the world's population, and 8.05 million active users in Portugal, comprising 78.5% of the population, social media platforms play a significant role in

shaping public opinion and discourse (*DataReportal, 2023*). OSL emerges as a powerful strategy for combating health misinformation by closely monitoring and analysing online conversations, providing valuable insights, and facilitating the dissemination of accurate health information to diverse audiences (*WHO, 2021b*).

While YouTube, Instagram, and Facebook currently dominate social media usage in Portugal, Twitter stands out as an effective platform for conducting OSL about health topics. With 328 million active accounts globally (1.90 million in Portugal), Twitter offers real-time updates, diverse perspectives, and a text-centric format that simplifies data processing (*DataReportal, 2023*). Its concise nature encourages users to share opinions, facilitating the capture of current sentiments and trends. The platform's interactive features, including hashtags and trending topics, provide convenient ways to track health-related discussions and engage with a wide range of viewpoints. Twitter usually serves as a central hub where discussions from various platforms converge, extending the reach of topics gaining traction on other platforms like Instagram, Facebook, or YouTube (*Gumaei et al., 2022*; *Ola & Sedig, 2020*; *Colditz et al., 2018*; *Hart et al., 2017*; *Suarez-Lledo & Alvarez-Galvez, 2021*). The prevalence of health misinformation on Twitter, combined with active participation from health professionals and organizations, makes it a valuable resource for accurate information and insights into public perception of health topics.

Crucially, the integration of diverse data sources, including both social media platforms (which offer a plethora of user-generated content) and traditional news outlets and magazines (widely regarded for their credibility), is paramount. PressReader (*PressReader, 2003*) emerges as a notable platform for accessing a wide array of news articles and publications, offering in-depth coverage and analysis on health issues, including NCDs affecting children. With its extensive collection of newspapers and magazines from around the world, PressReader provides a comprehensive view of global health trends and research findings. Its user-friendly interface and search functionalities enable efficient navigation through a wealth of content, making it a valuable resource for professionals and stakeholders interested in gaining insights into the prevalence, management, and impact of NCDs among children.

The motivation behind this article stems from its integral role within the Nursing School's research project at the University of Minho. The project is titled "Health promotion in primary school children and families in conditions of social vulnerability (BeE-school)—A cluster randomized study" (*Universidade do Minho, 2022*).

Through this project, there was direct interaction with children from vulnerable communities in Braga. This interaction revealed their limited access to daily health information. It also highlighted a common trend among children aged 6 to 10: frequent internet and social media use. This first-hand experience underscored the necessity for a comprehensive system to address health information dissemination among children, driving the need for this study. Additionally, the pursuit of sustainable development goals emphasizes the importance of reducing NCD mortality and promoting healthy lifestyles, especially among children (*WHO, 2022b*, *2023b*).

The kernel of this study is the development of an OSL system that systematically monitors and analyses social media discussions related to children's health behaviours,

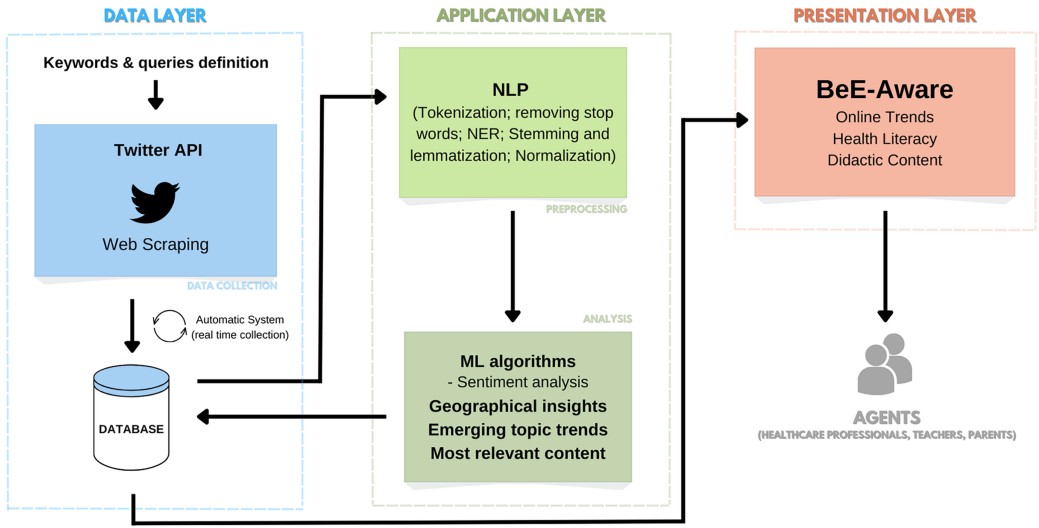

**Figure 1 Project pipeline: an OSL system for tracking trends related to NCDs and associated risk factors.** (https://www.flaticon.com/free-icon/twitter_733635?related_id=733579&origin=search).

particularly concerning NCD-related misinformation. By leveraging real-time data from Twitter and PressReader, this system aims to assist healthcare professionals in identifying misleading health narratives and promoting evidence-based health literacy interventions.

Creating an OSL system for tracking trends related to NCDs and associated risk factors in children provides numerous advantages, including early identification of health trends for proactive interventions, and facilitating focused efforts on specific areas like children's nutrition and exercise (*Braga et al., 2023*). Additionally, the analysis of publications and sentiments offers insights into public concerns, aiding health professionals in understanding diverse perspectives. This system has the potential to revolutionize social media interaction regarding NCDs and children's health. However, collaboration with influential agents such as governments, healthcare professionals, and educators is crucial for the effective dissemination of accurate information (*Bin Naeem & Kamel Boulos, 2021*).

In summary, Fig. 1 shows a schematic representation of the envisioned system pipeline. The work aims to achieve the following scientific and technical goals:

- Understand Twitter API functionality and its limitations.
- Investigate the impact of query formulation on data acquisition and evaluate different query strategies.
- Develop a robust system architecture for efficient data collection from social media, emphasizing graphical visualization.
- Establish a systematic pipeline for cleaning and processing acquired data to prepare it for analysis.
- Explore methodologies for sentiment analysis on collected data to understand expressed sentiments.
- Develop strategies for promoting health literacy within the data treatment process.

- Investigate automation of the entire back-end process to streamline data collection, cleaning, and processing.

## STATE OF THE ART

In recent years, researchers have increasingly focused on leveraging OSL to navigate the abundance of health-related information on social media platforms, particularly in light of the COVID-19 pandemic. This section presents a compilation of studies exploring OSL and its diverse applications in analysing health-related content.

One implementation was an advanced natural language processing (NLP) pipeline to sift through vast amounts of Twitter data and identify potential COVID-19 cases in the United States was proposed (*Klein et al., 2021*). By analysing tweets mentioning COVID-19 keywords, they employed sophisticated algorithms to detect patterns indicative of individuals self-reporting symptoms or potential exposure to the virus. This approach enabled them to contribute to the early detection and tracking of COVID-19 outbreaks through social media surveillance.

*Klein et al. (2021)*, developed a comprehensive taxonomy and methodology specifically tailored for extracting and categorizing relevant COVID-19 narratives from social media platforms (*Purnat et al., 2021*). By combining expert knowledge with Boolean string searches, they crafted a systematic approach to filter digital content related to COVID-19 and classify it based on its relevance to public health. This framework not only facilitated efficient content classification but also provided valuable insights for public health response planning and risk communication strategies.

In 2018 a Real-time Infoveillance of Twitter Health Messages (RITHM), an innovative framework designed to streamline the process of collecting and analysing data from Twitter for public health research purposes was introduced (*Colditz et al., 2018*). RITHM offers researchers a robust set of tools for data collection, storage, filtering, and formatting, aiming to maximize efficiency and effectiveness in subsequent data analysis. By providing a structured approach to data handling and analysis, RITHM enhances insights into public health trends and behaviours gleaned from social media data.

Recent studies have deeply expanded the role of OSL in identifying and countering health misinformation, particularly in relation to children's health and social media engagement. For example, *Sharevski & Loop (2023)* investigated how children interact with misinformation online, revealing that while they can recognize certain deceptive content, they often rely on digital assistants like Siri or their parents for verification (*Sharevski & Loop, 2023*). This highlights a crucial gap in media literacy, emphasizing the need for targeted interventions to improve children's ability to discern misinformation. Their study supports the current research by reinforcing the importance of OSL in analysing parent and educator discussions, which shape children's perceptions of health-related topics such as nutrition and physical activity. Additionally, *Garwood-Cross (2023)* explored one very intersting topic that is the increasing reliance of children and young people on social media for health information (*Garwood-Cross, 2023*). The authors underscore both the benefits and risks associated with user-generated content. This aligns with the present study's
objective of using OSL to track discussions on children's health across social platforms like Twitter. *Garwood-Cross (2023)* focused on the UK context, the methodologies employed in this study extend to a broader dataset, capturing a diverse range of conversations that influence children's health behaviours. By incorporating insights from various sources, including traditional news media and social media discussions, the study enhances the ability to identify prevailing narratives, assess misinformation trends, and develop strategies to promote accurate health information. Furthermore, *Papanikou et al. (2024)* conducted a systematic survey of IT approaches for tackling health misinformation in social networks, emphasizing the role of misinformation detection, fact-checking, and AI-driven classification methods (*Papanikou et al., 2024*). Their findings contribute to the methodological framework of the present study, reinforcing the necessity of integrating machine learning and NLP techniques into OSL systems. Over and above that, *Heyerdahl et al. (2023)* provided a critical perspective on the use of social listening in global health contexts, advocating for a more inclusive and context-aware approach (*Heyerdahl et al., 2023*). Their argument for grounding social listening methodologies in local cultural contexts aligns with the present study's approach, which seeks to capture nuanced discussions across different demographics and linguistic groups. By incorporating these recent contributions, this study enhances the scope and applicability of OSL in combating health misinformation, particularly in relation to children's health and NCD prevention.

Additionally, various online platforms, such as the WHO's EPI-WIN (*WHO, 2021a*) initiative and Symplur's Healthcare Hashtag Project (*Symplur, 2023*), actively employ OSL to address misinformation, particularly concerning the COVID-19 pandemic. These platforms utilize diverse data sources and automated categorization to track online discussions and provide insights into trending topics and public sentiments. Moreover, initiatives like Portugal's Polígrafo newspaper and the CovidCheck.pt website demonstrate efforts at both national and local levels to combat misinformation and promote reliable information dissemination.

Overall, the studies and platforms presented above offer valuable techniques and approaches that can enhance infodemic management systems. Despite their diverse research objectives, their methodologies and insights hold potential for integration into infodemic management strategies, providing flexible tools to combat health misinformation and manage information dissemination during public health crises.

These recent contributions collectively highlight the evolving landscape of OSL in health research, emphasizing the continuous development of NLP techniques and machine learning models for monitoring, classifying, and addressing health-related misinformation. Such advancements are particularly relevant for the present study, which aims to apply these methodologies to analyse online discourse surrounding children's NCDs and identify prevailing misconceptions that may hinder public health efforts.

The OSL platform was chosen for this study due to its ability to monitor public discussions on health topics, especially those related to children's NCDs, by analysing online content from diverse sources. While children aged 6–10 typically have limited smartphone access, the OSL approach focuses on capturing conversations from parents, educators, and healthcare professionals—key influencers in shaping children's health

behaviours. By analysing topics like nutrition, physical activity, and sleep patterns, the study seeks to understand how these issues are discussed online and to identify misinformation that could impact children's health.

In conclusion, while OSL proves invaluable for understanding and addressing health-related issues, navigating through vast amounts of unstructured data requires sophisticated tools and methodologies, including NLP and machine learning (ML) algorithms for sentiment analysis and semantic understanding.

## MATERIALS AND METHODS

The project devised a high-level architecture outlining the chosen software components and their interrelationships. The selected technologies for the project include Node.js, the Mongo Atlas database, Python in conjunction with Google Cloud Functions for the back-end server, and React.js for the front-end server. A distinctive feature of this system architecture is the strategic use of dual technologies for the back-end server. Google Cloud, supported by Python scripts and Cloud Scheduler, functions as the primary engine for data collection, cleaning, and processing. Simultaneously, Node.js plays a pivotal role in facilitating database connectivity and integrating with React.js.

The system architecture, represented in Fig. 2, is divided into four main components: data source; server back-end; data storage and website front-end.

All of the system's components and all of the work done up until the application's final state will be thoroughly discussed, but, in general, the system architecture of Be-E Aware is designed around these integral components:

- **Data source:** This component acts as the gateway to real-time data from the social media platform by leveraging the Twitter API. It collects tweets and associated data for subsequent processing.
- **Server back-end:** Using Google Cloud Functions as its backbone, the server-side back-end manages the tasks of data collection, cleaning, and processing. Google Cloud Functions enable efficient handling of data collection and cleaning processes, while Node.js serves as the intermediary connecting the database to the front-end server.
- **Data storage:** The robust data storage solution, MongoDB, efficiently manages the structured data gathered from Twitter. It ensures reliable data accessibility and integrity, enabling seamless storage and retrieval of data within the system.
- **Website front-end:** The user-friendly interface of the website, designed for accessing and interacting with processed data, was developed using React.js. This interface empowers users to intuitively explore trends, insights, and health-related information with ease.

### Data collection

The project selected Twitter API as its primary data source for several reasons. Firstly, Twitter is a widely recognized social media platform with a diverse user base engaging in

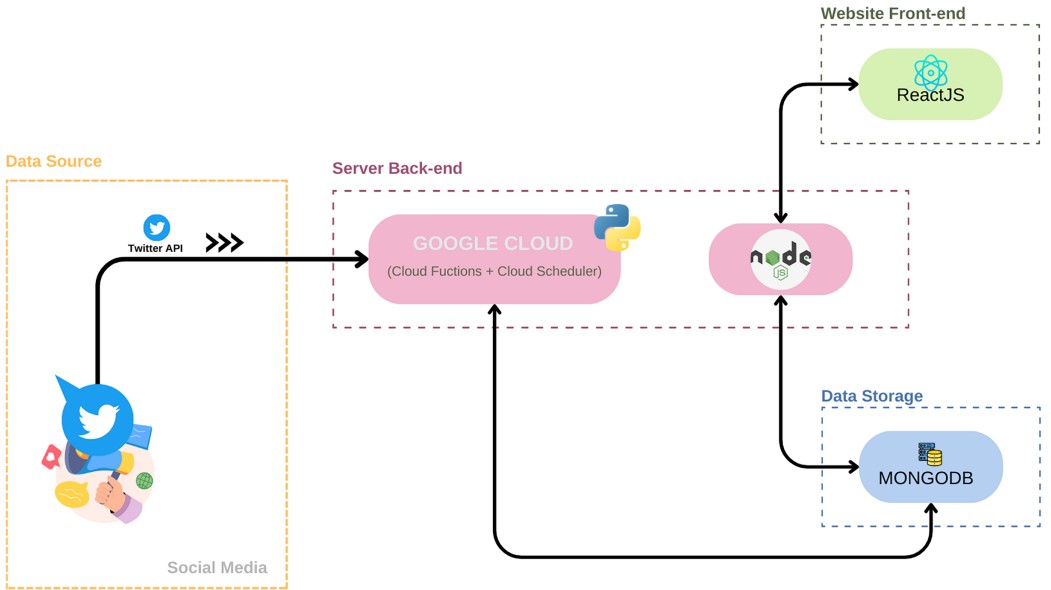

**Figure 2 System architecture: a visual map of the Be-E Aware health trend monitoring platform.** (https://www.flaticon.com/free-icon/twitter_3670151?term=twiter&page=1&position=2&origin=search&related_id=3670151. https://www.flaticon.com/free-icon/publicity_17524829?term=social+networks&page=1&position=57&origin=search&related_id=17524829. Python–https://www.flaticon.com/free-icon/python_5968350?term=python&page=1&position=6&origin=search&related_id=5968350. NodeJS–https://www.flaticon.com/free-icon/nodejs_919825?term=nodejs&page=1&position=1&origin=search&related_id=919825. React–https://www.flaticon.com/free-icon/physics_1126012?term=react&page=1&position=3&origin=search&related_id=1126012. Mongo–https://www.flaticon.com/free-icon/database-storage_2906274?related_id=2906206&origin=search).

discussions on various topics, including health. Secondly, the project's emphasis on monitoring and responding to emerging health-related trends aligns seamlessly with Twitter API's capability to provide developers with real-time data access. Thirdly, the open framework of Twitter's API allows for flexible crafting of specific data collection queries, facilitating precise targeting of relevant content. Lastly, the public availability of Twitter data enhances the project's transparency and accessibility, adhering to ethical data usage practices.

While Twitter API supports academic research and application development for free, it comes with limitations such as rate limits (300 requests per 15 min) and access restricted to public tweets. Despite the potential for real-time data acquisition, the project opted for daily updates to efficiently meet its goals, considering the API's rate limits. This strategy addresses concerns about identifying pertinent tweets in real-time, as daily updates provide information that approximates real-time conditions, ensuring effective data collection within the API's constraints.

Selecting appropriate keywords is a vital and intricate task in data collection, especially when aiming to comprehend complex subjects like NCDs and their associated risk factors. The significance of keywords lies in their role as gatekeepers to the vast realm of digital

information, particularly in this text-centric data context. The selection process followed a structured approach based on multiple criteria:

- **Expert consultation:** A panel of health experts, including epidemiologists and public health professionals, was consulted to identify essential terms related to NCDs and child health. Their expertise ensured that the selected terms were both medically and contextually relevant.
- **Literature review:** Scientific publications, policy reports, and global health databases (*e.g.*, WHO, CDC) were analysed to identify commonly used terms in discussions about child health, lifestyle habits, and NCD risk factors.
- **Data-driven expansion:** To enhance keyword coverage, we utilized the NLTK WordNet library and the Related Words platform to identify synonyms and semantically related terms, ensuring broader data retrieval.
- **Preliminary data testing:** An initial dataset was collected to evaluate keyword effectiveness. Keywords generating a high volume of irrelevant data were either refined or excluded.
- **Filtering for context:** To improve relevance, queries included additional terms specifically related to children (*e.g.*, "kids," "child," "minor"). Retweets, quotes, and replies were also excluded to prioritize original content.

Through this multi-step approach, a refined set of 116 unique terms was established, balancing inclusivity and precision in data collection. However, during individual term searches, a significant amount of irrelevant information was collected, prompting the need for control measures. To filter this data, various experiments were conducted, initially extracting tweets with combinations of two out of the 116 predetermined keywords. Semantic similarity between keyword pairs was measured using the spaCy NLP library, but this approach proved ineffective. A refined strategy involved extracting concepts associated with younger age groups to improve data filtration precision.

The study involved analysing 103 distinct queries spanning 15 primary topics related to child health and behaviour, such as "lifestyle habits", "substance abuse", "diets", "nutrition", "physical exercise", "sleep", "screen time", "diabetes", "hypertension", "cardiovascular disease", "cancer", "mental health", "noncommunicable diseases", "respiratory diseases", and "school". For the final dataset, each of the 103 queries was refined to include the original predetermined keyword, terms specific to children (kids, child, minor …), and a filter to exclude retweets, quotes, and replies, thereby retaining only original tweets (*Peixoto et al., 2025*). This method, although resulting in some irrelevant data, aimed to prioritize the acquisition of an extensive dataset for a comprehensive analysis of child-related topics.

The study collects diverse data attributes from Twitter, including Tweet ID, text, creation date, language, user information, multimedia information, entities, and public metrics. User information, like verification status, helps identify influential participants. The location attribute provides insights into the geographic concentrations of discourse, while links and multimedia files enhance the depth of the tweet *corpus*. Retrieving

public metrics, such as engagement and retweet counts, aids in identifying influential content. Data extraction from Twitter's API involves making requests with parameters like query, start and end times, and maximum results. Due to API limitations, the daily routine extracts tweets from the past 24 h. Python's *requests* library is used for extraction, and the data is stored in a Mongo Atlas database in a structured dictionary format.

Initially, tweets were indexed based on ID numbers, but due to users repetitively posting identical tweets, a revised indexing scheme was introduced. This aims to retain the original tweet with the highest user engagement while accommodating multiple entries for users disseminating the same content. This approach prevents redundant storage of identical tweets and conserves computational resources.

## Data cleaning and processing

Twitter data, rich in potential insights, requires thorough preprocessing before meaningful analysis. Filtering mechanisms like "health_filter" and "pos_filter" were implemented to exclude irrelevant information. Some selected keywords lacked a direct association with health, introducing potential noise into the dataset (*e.g.*, "school" and "television"). As a solution, the "health_filter" was introduced to assess tweets and identify matches with a predefined list of health-related keywords. To handle nuanced language usage, the "pos_filter" function was developed using the *spaCy* library. This function specifically addressed instances where users used "minor" as an adjective and "kid" as a verb, requiring specialized management to maintain dataset accuracy and relevance to research goals.

Cleaning procedures for tweets involved several steps (*Bopaiah, 2022*; *Pradha, Halgamuge & Vinh, 2019*):

- **HTML entities removal:** Removed entities like "<", ">", and "&" using regular expressions.
- **Link extraction:** Extracted and stored links separately for reference.
- **Standardization:** Transformed abbreviations and shortened forms for coherence.
- **Hashtag and mentions handling:** Preserved hashtags and mentions for analysis.
- **Lowercasing:** Converted all text to lowercase for uniformity.
- **Stop-word removal:** Eliminated common words using *NLTK* stop-words.
- **Punctuation removal:** Removed all punctuation marks.
- **Laughter normalization:** Normalized repetitive laughter instances.
- **Shortened repeated characters:** Truncated consecutively repeated characters for readability.
- **Emoji handling:** Translated emojis into text equivalents.
- **NER (named entity recognition):** Identified and tagged named entities for further analysis.
- **Tokenization:** Divided tweets into tokens (individual terms) for analysis.

**original_tweet**

🌍 Today is World Food Safety Day!

Healthy eating is important for everyone to lower the risk of developing #NCDs such as #T2D and #hypertension and is crucial for the care and management of these conditions.

▶️ Check out our infographic on healthy eating: digicare4you.eu/content/digica...

**cleaned_text**

"emoji globe showing Europe–Africa emoji today world food safety day healthy eating important everyone lower risk developing ncds td hypertension crucial care management conditions  emoji right arrow emoji check infographic healthy eating"

**tokens**

["today", "world", "food", "safety", "day", "healthy", "eating", "important", "everyone","lower", "risk", "developing", "ncds", "type–2-diabetes" "hypertension", "crucial", "care", "management", "conditions", "check", "infographic", "healthy", "eating"]

**Figure 3** **Example of cleaned tweet after the data cleaning process.**

Python packages like *re*, *NLTK*, and *pysentimiento.preprocessing* (*Pérez, Giudici & Luque, 2021*) were used for efficient data cleaning. Numeric values were retained in English tweets related to type 2 diabetes. Tweets in other languages were translated using the "googletrans" package before cleaning. At the end of this process, obtaining two text types for future analysis was possible, as depicted in Fig. 3.

The goal was to achieve a balanced representation of sentiments within the dataset. Ultimately, 100 test tweets were categorized into three distinct sentiment classes: 33 positive, 33 negative, and 34 neutral tweets. This structured distribution facilitated comprehensive evaluation and comparison of sentiment analysis models, including TextBlob, VADER (*NLTK*), BERTweet (*Pérez, Giudici & Luque, 2021*), and DistilBERT (*Sanh et al., 2019*), following rigorous data cleansing procedures.

For BERTweet, the "finiteautomata/bertweet-base-sentiment-analysis" tokenizer and pre-trained model were used. DistilBERT employed the "distilbert-base-uncased-finetuned-sst-2-english" model and the pipeline from the "Transformers" package. In the case of TextBlob and VADER, sentiment polarity was categorized as positive, neutral, or negative based on specific polarity value ranges (*Hota, Sharma & Verma, 2021*).

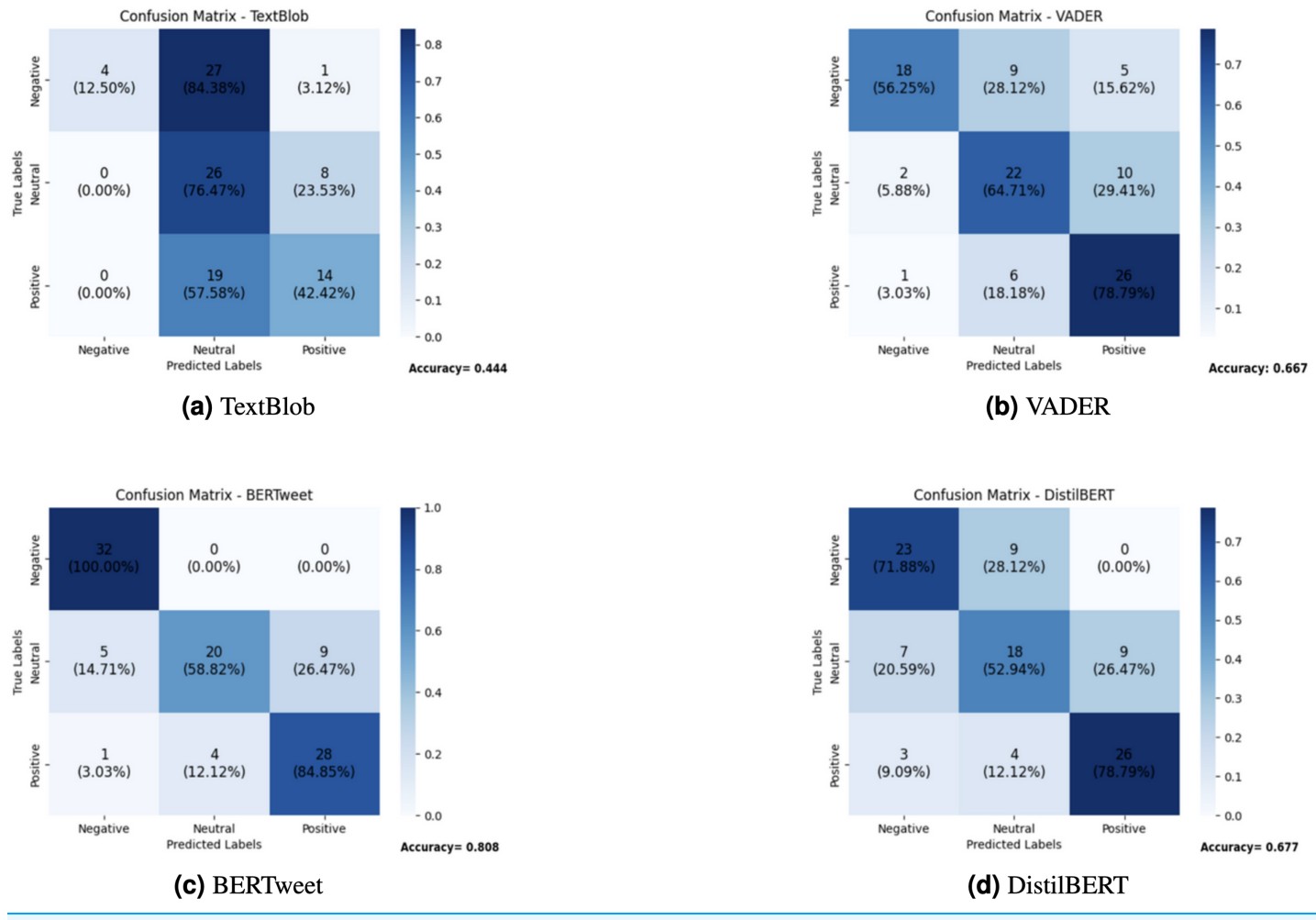

**Figure 4 Comparison of sentiment analysis models: confusion matrices.** (A) TextBlob. (B) VADER. (C) BERTweet. (D) DistilBERT.

The effectiveness of each sentiment analysis model on the 100 test tweets was evaluated using correlation matrices and accuracy values, as depicted in Fig. 4.

TextBlob's performance was subpar, predominantly categorizing tweets as neutral, indicating a limitation in discerning nuanced sentiments. Adjustments to the neutrality threshold did not significantly enhance its accuracy. With a precision of 0.650 and recall of 0.444, TextBlob struggled to correctly classify sentiments, particularly negative ones. VADER, while highlighting specific qualities, misclassified neutral tweets as positive, a concerning issue for health analytics, potentially leading to an overestimation of positive sentiment. Its precision of 0.666 and recall of 0.646 indicate moderate performance, though its F1-score suggests inconsistency in classification. DistilBERT demonstrated performance comparable to VADER but struggled with accurately identifying neutral tweets, achieving a precision of 0.672 and recall of 0.677. In contrast, BERTweet emerged as the standout performer, boasting an accuracy rate exceeding 80%, a precision of 0.811 and recall of 0.808, leading to an F1-score that surpassed all other models. In this research,
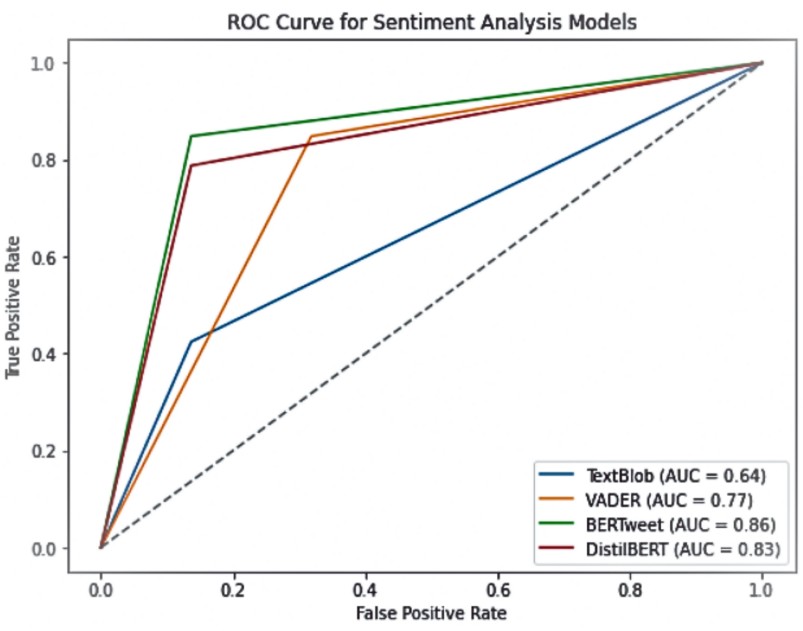

**Figure 5  ROC curve for sentiment analysis models.**   

the decision to utilize a pre-trained Sentiment Analysis model, such as BERTweet, was grounded in its proven performance, extensive training on diverse datasets, and the ability to generalize effectively in the dynamic context of social media content. This approach not only saved valuable resources but also ensured efficient capture of sentiment nuances in health-related tweets, with BERTweet achieving a notable 100% accuracy in identifying negative sentiments in the test sample.

To further assess model performance, we evaluated their receiver operating characteristic (ROC) curves, measuring the true positive rate against the false positive rate. The area under the curve (AUC) values provide a robust metric for comparing classification performance. As shown in Fig. 5, BERTweet demonstrated the highest AUC score (0.86), followed closely by DistilBERT (0.83), while VADER and TextBlob performed comparatively worse with AUC values of 0.77 and 0.64, respectively.

The geolocation challenge arose from users providing fictional locations. Initially, Google's geocode API was used, but it became time-bound due to payment-related restrictions. Overcoming this, Web Scraping from the GeoNames website (*GeoNames, 2023*) improved dataset accuracy by acquiring location data. Utilizing regular expressions and the *PySentimiento* model, tweets expressing concern or doubts were identified based on the predominant emotion of fear. A scoring system prioritizing relevance and significance considered user interactions (likes, retweets, views), identifying high engagement as indicative of notable trends. Verified users' tweets, owing to their influence, were given added weight in shaping public discourse.

After data processing, information selection for the application followed three paths: general stats, topic-specific analysis, and link-containing tweets treatment. General stats covered tweet count, engagement, sentiment, and geographic distribution. Research topics,

ranked by frequency, considered hashtags, and tweet tokens for trend identification. Opting for tokens in trend analysis preserved language nuances, revealing insights into distinct discussions, like varied discussions around different diabetes types. The established scoring system facilitated the identification of the top 10 tweets across diverse categories, including global impact, linked multimedia, verified user tweets, and expressions of concerns or doubts. Each category served a unique purpose. The top 10 global tweets signify exceptional engagement and resonance, showcasing their relevance and widespread impact. Conversely, the top 10 tweets with links or multimedia underline the role of multimedia in enhancing tweet visibility and engagement. Verified users, with their influential status and likely greater credibility, form a distinct category, given their substantial impact on public discourse. The concerns and doubts category identifies tweets expressing ambiguity about health-related topics, emphasizing areas requiring clarification. Similar to general statistics, a comparable method was employed for topic-specific health statistics.

To enhance their significance, tweets with links were singled out for special treatment. After reviewing the data, it was evident that many of these links led to informative articles, studies, or news relevant to the research. Recognizing their value, a decision was made to categorize and manage them separately, potentially for a health literacy or news section within the app. To achieve this, a filtering process was applied to select links with titles and descriptions containing keywords related to research, scientific investigations, and advancements. Keywords like "research", "article", "study", "discovery", and reputable medical publication sites were included. Additionally, the term "government" was added to include authoritative health-related publications. This comprehensive approach ensured the collection of accurate sources and relevant health data, with the filtered links archived alongside corresponding topics and associated scores.

## System automation

The vital system automation component in this project played a crucial role in ensuring the continual and efficient execution of tasks related to data acquisition, cleaning, and processing. Google Cloud services, specifically Cloud Scheduler and Cloud Functions, were employed for this purpose. The architecture of the automation system is illustrated in Fig. 6. Cloud Scheduler functions as a timekeeper, triggering PUB/SUB to initiate the automation process. PUB/SUB, functioning as a communication channel, alerts Google Functions to execute specific tasks. These functions, in turn, activate a Python script serving as a central processing unit responsible for collecting and processing data. The processed results are then stored in Mongo Atlas database, serving as a data repository.

Daily, precisely at 12:01 AM, Python scripts engineered to execute a sequence of pivotal operations were triggered, encompassing tasks such as data collection, cleaning, processing, and analysis. This commencement signified the initiation of a novel data processing cycle. Central to this process was the adept management of data within Mongo Atlas database, constituting a fundamental facet of the overall operation. As a constituent of this procedure, data from the preceding day underwent removal from the database, ensuring the precision and timeliness of the information presented to users.

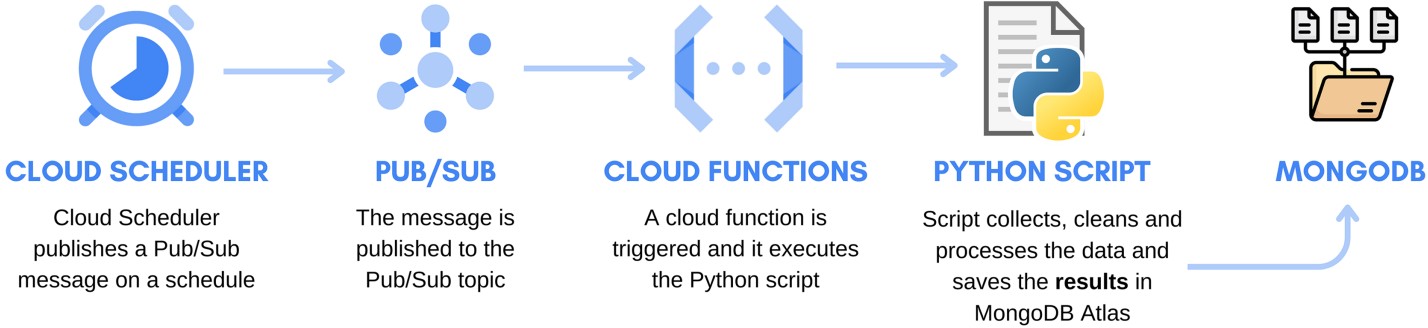

**Figure 6 Automation system architecture. Adapted from** *Costa (2019)*. (https://www.flaticon.com/free-icon/python_5968350?term=python&page=1&position=6&origin=search&related_id=5968350. https://www.flaticon.com/free-icon/folder_3025405?term=mongodb&page=1&position=45&origin=search&related_id=3025405).

Simultaneously, the newly processed data, embodying the latest insights and statistics, was integrated into the database, facilitating user access to the most up-to-date information.

**PressReader**

To comprehend whether the most discussed topics on social media align with those discussed in the press, we employed the same research keywords mentioned in the *Data Collection* subsection. These keywords, carefully selected to capture relevant discourse, were filtered specifically for June 7, 2023, providing a snapshot of discourse on that particular day. By comparing the frequency of these keywords on PressReader with that on Twitter, we aimed to discern any potential discrepancies or similarities in the topics garnering attention across these two distinct media platforms.

The selection of June 7, 2023, as the date for filtering the research keywords serves as a temporal anchor for the Twitter results, ensuring a consistent time frame for comparison. This allows for a more accurate juxtaposition of the frequency of topics discussed on both platforms, offering insights into potential temporal trends or synchronicities in public discourse between social media and traditional press outlets.

## RESULTS

An OSL platform, developed using React.js, was designed to monitor and analyse conversations and trends related to children's health across the Twitter platform. This platform enables users to gain insights into public sentiments, discussions, and emerging issues concerning children's health in real-time. It is imperative to underscore that the genesis of this platform primarily aligns with the exigencies of the scientific and research community, as well as health professionals possessing adept knowledge in the domain, enabling proficient interpretation of information.

The main page provides a comprehensive overview, while a dedicated page meticulously curates detailed daily statistics. Furthermore, an exclusive page is devoted to statistics pertinent to each research topic, demonstrating the platform's adaptability. An additional page is designated for presenting a carefully curated compilation of links, news, and

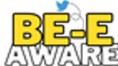

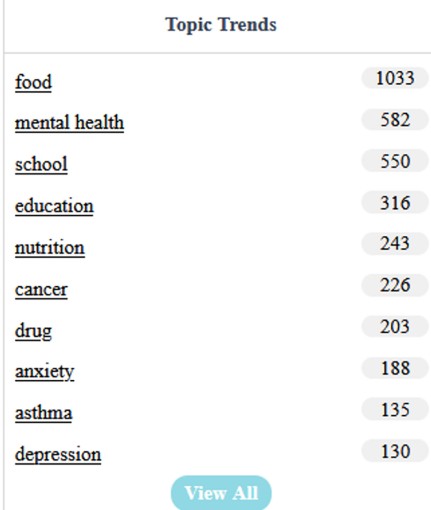

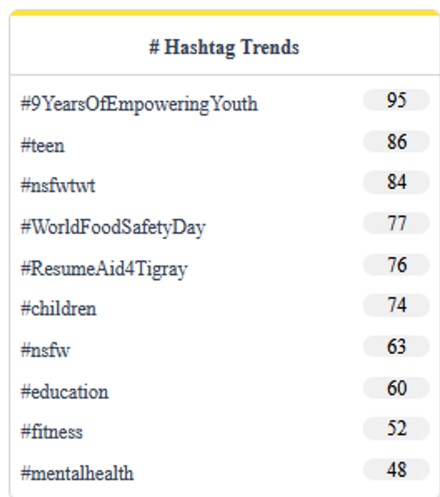

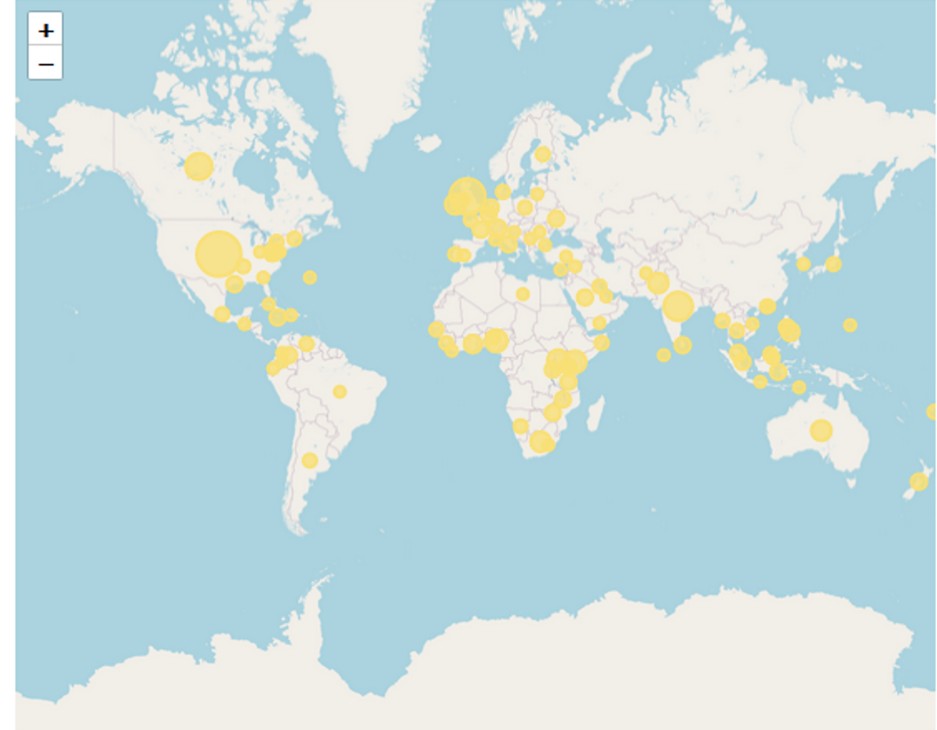

**Figure 7** Be-E Aware main page.

academic articles relevant to our research topics. It is imperative to emphasize that this section transcends the mere dissemination of information; rather, it is intricately crafted to contribute to the augmentation of health literacy among users through the best scientific evidence at the moment.

Users can find an overview of the daily Twitter conversation and an illustrative explanation of the app's primary function on the main page (Fig. 7). These have access to a
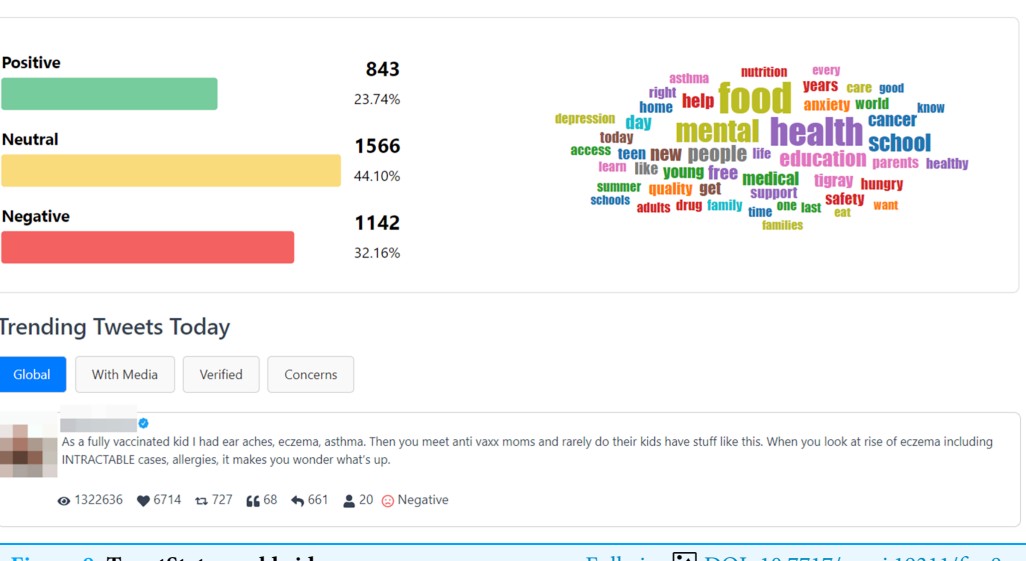

**Figure 8  TweetStats worldwide page.**

carefully chosen list of the hashtags and themes that are now trending on Twitter. Furthermore, the main page prominently showcases the cumulative engagement metrics for the day, which include the number of likes and tweets among other indicators. There is also an interactive global map that provides information on the geographic distribution of health-related conversations.

The "TweetStats Worldwide" page, depicted in Fig. 8, serves as a complementary resource to the main page, furnishing users with supplementary statistical insights into the complete dataset. Within this section, users can delve into a sentiment analysis, accompanied by a word cloud that accentuates the most frequently used words in the tweets. Furthermore, the page presents the "Top 10 Trending Tweets of the Day" across four distinct categories: Global Trends, Tweets with Media, Tweets from Verified Users, and Tweets Addressing Concerns. This page is designed to provide users with a more extensive comprehension of prevalent discussions and the content encapsulated within the dataset.

To ensure the credibility of the news sources, the system specifically selects tweets containing links to external content. These links were then filtered to prioritize trusted and authoritative sources, such as prominent health organizations, government websites, and reputable academic and media outlets. By focusing on these reliable sources, the platform ensures that the information users encounter is both accurate and relevant for improving health literacy. The "Perspectives and Updates" page functions as a specialized central hub for navigating a meticulously curated assortment of URLs related to news, articles, and research within the purview of the research topics. Each link is systematically categorized to denote the specific topic it addresses, and they are arranged in order of relevance value. This layout facilitates seamless exploration for users, allowing them to effortlessly delve into pertinent external resources (Fig. 9).

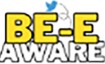 BE-E AWARE    TweetStats Worldwide    Health Topics    Perspectives and Updates

## Perspectives and Updates

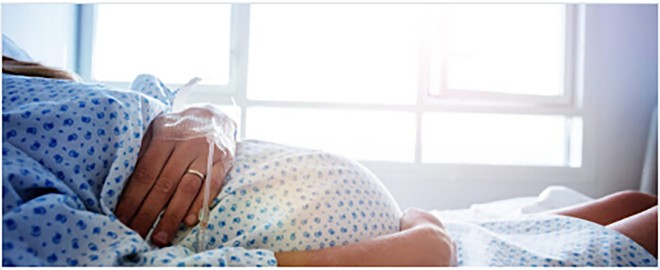

**NH moms, children among participants in study linking PFAS exposure to obesity risk – New Hampshire Bulletin**
A new National Institutes of Health study suggests that prenatal exposure to PFAS is linked to slightly higher body mass index and obesity risk in children. Some of the data used came from...
newhampshirebulletin.com

Related Topics: obesity

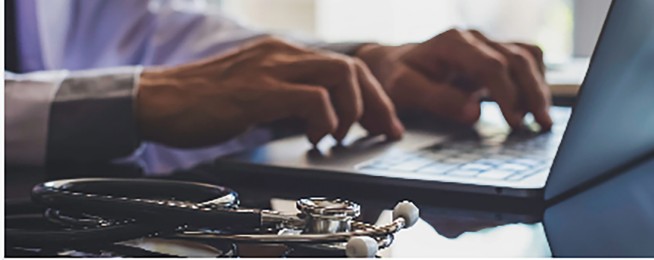

**Using telehealth to optimize patient care in pediatric type 1 diabetes mellitus**
A recent study looked at how patients with type 1 diabetes used telehealth during the pandemic, and how it has influenced their preferred method of health care.
contemporarypediatrics.com

Related Topics: diabetes

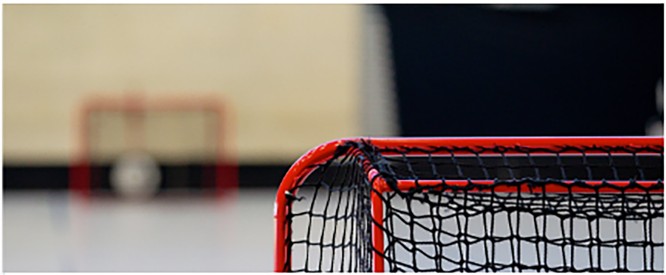

**Youth sports have positive impact on mental health, study says**

The study showed that participation in youth sports led to an increase in self-esteem.

turnto23.com

Related Topics: mental health

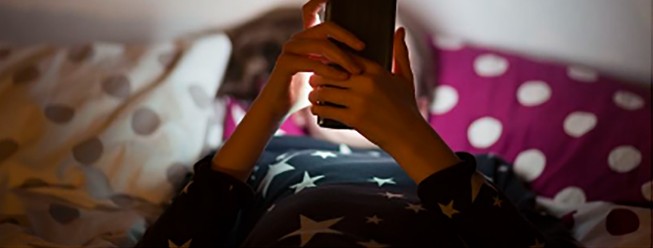

**Social media presents 'profound risk of harm' for kids, surgeon general says, calling attention to lack of research | CNN**
There's not enough evidence to determine whether social media is safe enough for children and adolescents when it comes to their mental health, according to a new advisory from the US...
cnn.com

Related Topics: mental health, social media

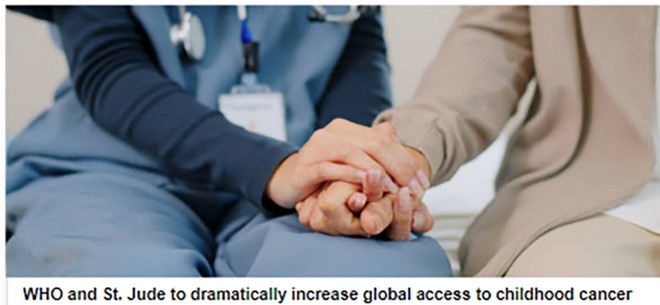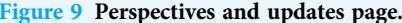

WHO and St. Jude to dramatically increase global access to childhood cancer

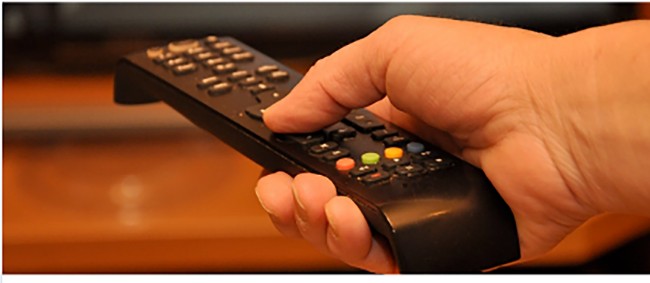

Norway to ban junk food adverts aimed at kids

**Figure 9  Perspectives and updates page.**

While general statistics offer valuable insights into a broad spectrum of health-related discussions, their inclusivity poses a challenge in pinpointing specific topics and trends within the dataset. To overcome this challenge, a specialized approach employing multiple
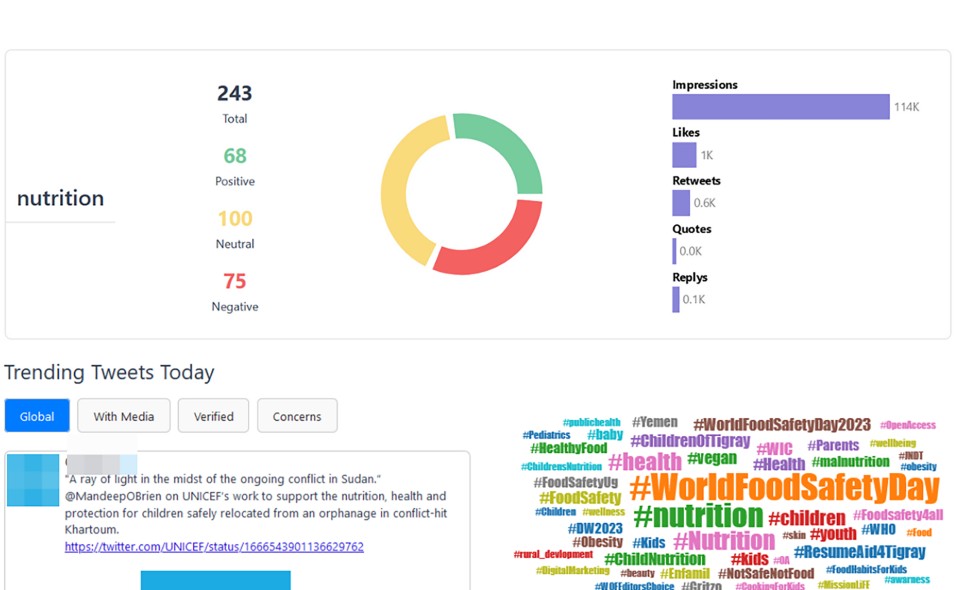

## Trending Tweets Today

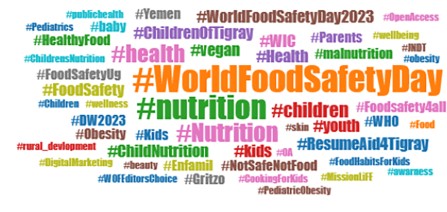

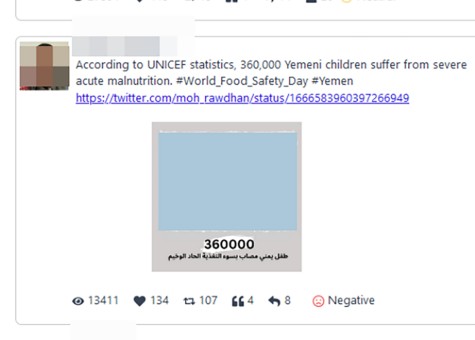

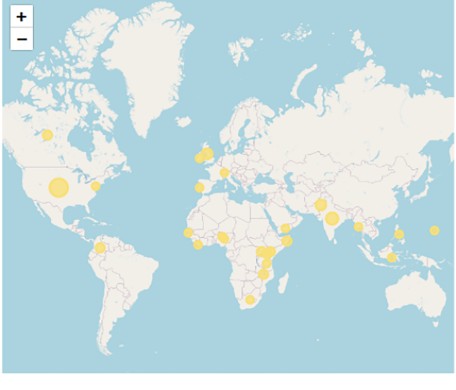

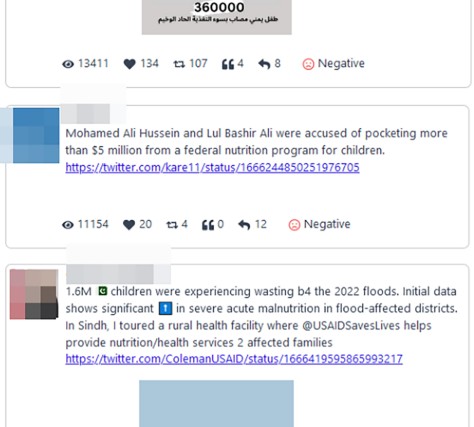

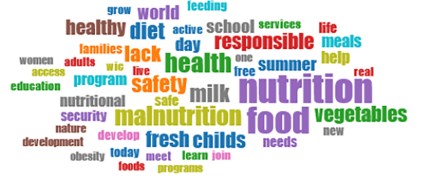

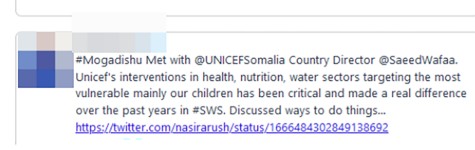

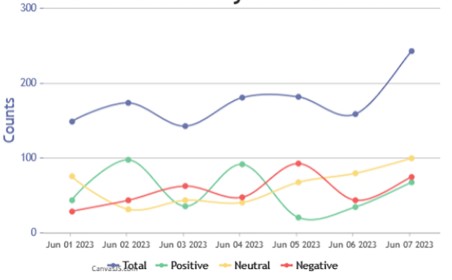

**Figure 10 Nutrition statistics page.**

(a) Frequency of topics in PressReader          (b) Frequency of topics in Twitter

**Figure 11** **Comparative overview of topic frequencies.**

pages has been adopted to analyse and present data pertinent to specific research topics. This targeted strategy enables a more precise examination of discussion trends and emerging topics within each distinct health category. Users gain access to a comprehensive list of all research topics, accompanied by information on the number of tweets published about each topic.

Within each specific topic, users encounter data of the same nature as found in the "TweetStats Worldwide" section but tailored exclusively to the topic under consideration, as can be seen in Fig. 10. Furthermore, each topic-specific page incorporates an informative graph illustrating the sentiment analysis over the past 7 days.

## PressReader and Twitter comparison

Analyzing Fig. 11, it is evident that Twitter displays a significantly higher frequency of discussions regarding research topics compared to PressReader. This finding highlights a substantial gap in the prevalence of discourse between the two platforms, indicating a greater inclination for topic engagement within the Twitter-sphere. This observation resonates with the inherent characteristics of each platform; whereas PressReader demands detailed textual content, Twitter encapsulates discussions within concise sentences.

Regarding the thematic content, a notable convergence is observed across both platforms, particularly pertaining to mental health-related topics such as "mental health", "depression", and "anxiety". Moreover, themes such as "cancer", "food", and "drug" also manifest commonalities between PressReader and Twitter. It is worth noting that this analysis considered only the top 10 topics, revealing that approximately 60% of these top topics overlap between the two platforms. This suggests a considerable degree of thematic

congruence in the treatment of subjects across both traditional press and social media domains.

## DISCUSSION

In the initial stages of this process, the proof of concept (PoC) was effectively established, showcasing operational functionality. However, a significant obstacle arose in June 2023 when the application reorganization by the new proprietor of X (Twitter's new name) led to substantial modifications in the API definitions. This modification resulted in the termination of free access to Twitter's Academic API, making data acquisition prohibitively expensive. Consequently, the accessible dataset was limited to records from June 7, 2023.

It is imperative to emphasize that the technical infrastructure of the project was systematically developed, except for the final testing and validation phases, which were pending. Despite their pending status, these phases were executed iteratively throughout the project to enhance the quality and integrity of the data presented to the target audience. Regrettably, financial constraints compelled the discontinuation of new data acquisitions, thereby restricting the analysis of the final results to the last day of data collection.

Data collection commenced on June 7, 2023, at 00:00 and concluded at the same time on June 8, 2023. Post-collection, a dataset of 3,551 unique tweets underwent meticulous cleaning and processing. This dataset garnered a total of 7,488,962 views, 78,077 likes, and 23,866 retweets.

Initially, it became evident that the data collection window on June 7, 2023, coincided with World Food Safety Day 2023. Notably, topics related to "food" dominated discussions on that day, constituting a total of 1,033 tweets. Furthermore, the prevailing discourse broadened to include discussions on "mental health", "school", "education" and "nutrition", offering valuable contextual insights into the public discourse on that particular day.

In analyzing the general statistics, it was found that approximately 44.10% of the collected tweets were categorized as neutral, with negative sentiments surpassing positive sentiments at 32.16% and 23.7%, respectively. This disparity underscores the prevalence of negative sentiments in discussions related to child health issues on the specified day. A closer examination of the dataset revealed that among the top 10 tweets with the highest global engagement, eight exhibited a noticeable negative sentiment, while the remaining two were classified as neutral.

A salient example in this group is the top-ranked tweet, which spread misinformation about vaccination, suggesting a potential link between vaccinations and health problems like earaches, eczema, and asthma in children (Fig. 12). Such unverified claims pose a significant public health risk by potentially discouraging parents from vaccinating their children. Addressing this and misleading information is crucial not only for maintaining vaccination coverage but also for reducing children's vulnerability to preventable diseases, highlighting the imperative need for effective communication on this topic.

Furthermore, another tweet within the top 10 referenced an alleged link between COVID-19 infection and long-term consequences in pediatric patients, including potential

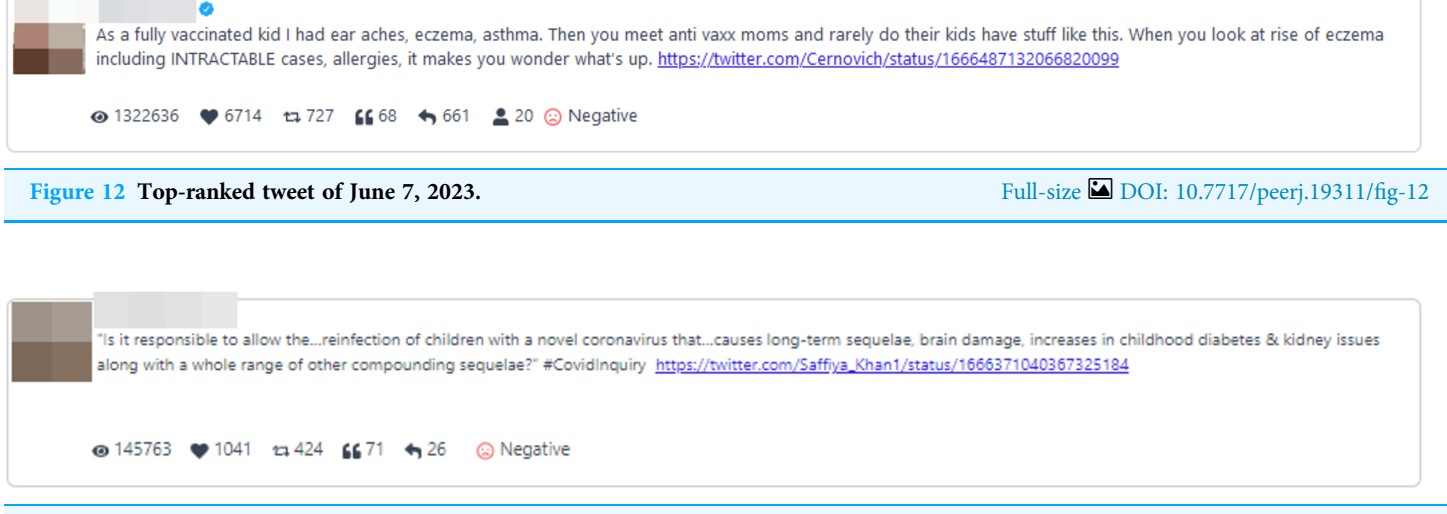

**Figure 12 Top-ranked tweet of June 7, 2023.**

**Figure 13 Ninth tweet of the top 10 global tweets of June 7, 2023.**

outcomes such as brain damage, heightened susceptibility to childhood diabetes, and renal complications (Fig. 13). Despite the speculative nature of these claims, they hold the potential to induce fear and uncertainty among parents whose children have previously been diagnosed with the disease. This underscores the imperative to debunk and provide clarification on such assertions.

The comparison between the top 10 tweets and those with multimedia attachments reveals minimal disparity, emphasizing the importance of incorporating visuals for increased engagement. Healthcare professionals can leverage multimedia to counter misinformation effectively. Additionally, the examination of the top 10 global tweets *vs.* verified users shows a slight improvement in content relevance. However, an unexpected increase in misinformation was noted in the posts, notably a tweet incorrectly stating that being transgender is a mental illness. The concerns/doubts section features valuable tweets expressing worries about child vaccination, mental health, and obesity.

While the data cleaning and filtering processes were conducted meticulously, it is important to acknowledge that, despite rigorous procedures, there may still be instances where tweets not directly relevant to the research question are present in the dataset. Such situations are inherent in the dynamic landscape of social media data collection. However, it is crucial to understand that the platform's effectiveness does not solely depend on eliminating irrelevant tweets. Instead, its primary strength lies in its ability to identify and analyze emerging trends in the population's discourse. The sentiment analysis also plays a crucial role in this study, providing healthcare professionals with a unique perspective to examine public opinions and concerns related to various healthcare themes. This perspective enables informed initiatives and responses to public concerns, ensuring that health information and services align with the population's sentiments.

This phenomenon is especially noticeable when a specific issue gains significant public attention, resulting in a surge of related tweets and increased user engagement. To

illustrate, consider a hypothetical scenario where a prominent influencer with a substantial online following raises concerns about the safety of commercial baby food brands. The influencer's message, emphasizing harmful chemicals and toxins in these products and advocating for homemade baby food, would likely trigger a wave of tweets discussing "baby food", "harmful chemicals", and "child development". This surge in related tweets would naturally elevate these topics to the forefront of trending discussions on the platform.

The "Perspectives and Updates" page features links to highly relevant articles for users, particularly parents, and teachers, covering topics such as the positive impact of youth sports on mental health, PFAS exposure's link to obesity risk, and the risks of social media on children's well-being. Despite commendable content, limitations in consistent information presentation arise due to variations in website structures and web scraping complexities. Occasional inclusions of content less aligned with the primary focus are expected, given the automated processes' reliance on keyword matching and contextual interpretation.

The findings from comparing the frequency of topics between PressReader and Twitter shed light on the distinct dynamics of discourse between traditional press and social media platforms concerning research topics. Twitter emerges as a prominent platform for discussions, characterized by its succinct nature and high frequency of engagement. In contrast, PressReader, while still relevant, appears to host fewer discussions on research topics, likely due to its emphasis on detailed textual content.

The thematic analysis reveals a noteworthy overlap in the thematic content discussed on both platforms, particularly in domains such as mental health, cancer, food, and drug-related topics. This convergence suggests that despite differences in platform structure and user demographics, there is a shared interest and concern regarding certain research topics across traditional and social media platforms.

## Research strengths and limitations

Through the analysis of the results in the previous section, it is evident that all the objectives outlined were successfully achieved.

The research gains strength from its diverse team and collaborative approach, enhancing credibility and effectiveness. Professionals from various fields, including health sciences, contribute diverse perspectives and knowledge, while jointly defining search topics, streamlining processes, and boosting credibility. Focusing on preventing NCDs in children aligns with global health priorities, demonstrating a significant strength. Additionally, the developed system's ability to swiftly identify prevalent topics proves crucial in the digital age, where online discourse shapes public opinion. This trend analysis feature allows healthcare professionals to grasp dominant themes from daily online conversations. Moreover, the app's significance grows during moments of heightened attention to specific health issues, engaging healthcare professionals in ongoing conversations and fostering trust as a reliable source of health-related knowledge in the

digital sphere. Furthermore, the research's use of PressReader to verify if Twitter information aligns with media sources adds a layer of reliability, enhancing the research's credibility.

Despite the numerous strengths inherent in the research, certain limitations should be acknowledged. Firstly, the focus on English-language tweets introduces potential bias and hinders generalizability, as non-English-speaking populations with distinct health-related conversations on social media may be under-represented. Secondly, relying solely on Twitter as the source of social media data may limit the comprehensiveness of the collected data, not fully capturing the diversity of online health-related discourse. Lastly, the generalizability of the application's conclusions may be constrained by the lack of regional organization, as different demographics and contexts could produce varying responses and results. Recognizing these limitations is crucial for a nuanced understanding of the study's scope and applicability.

Although this study does not directly compare the OSL system with traditional health monitoring methods, future research could incorporate comparative analyses with conventional surveillance tools to assess effectiveness in misinformation detection and public engagement.

Be-E Aware aligns with the principles and objectives exemplified by the WHO's Early AI-supported Response with Social Listening (EARS) (WHO, 2021a) and Symplur, aiming to integrate advanced technologies for early response and comprehensive social media monitoring in the realm of healthcare. EARS stands out for its extensive data sources, multilingual capabilities, and robust algorithm for categorizing data into various conversation topics. On the other hand, Be-E Aware excels with its comprehensive coverage of NCDs and associated risk factors, presenting data in a format conducive to human interpretation. Be-E Aware also incorporates sentiment analysis, providing deeper insights into public concerns, a feature lacking in EARS. Additionally, Be-E Aware addresses the challenge of punctuation symbols in topic keywords, a detail overlooked by EARS, which can compromise result relevance (Fig. 14). Overall, both platforms contribute significantly to health-related social media analysis with distinct features and capabilities.

SYMPLUR's healthcare hashtag project (Symplur, 2023) is akin to Be-E Aware in structure, connecting various users, including patient advocates, caregivers, doctors, and healthcare providers, to relevant conversations and communities. This makes it a highly pertinent tool in the healthcare domain. SYMPLUR distinguishes itself by providing real-time information and an extensive coverage of topics. However, its emphasis is on the identification of the most recent tweets and users with the highest engagement on a specific topic. In contrast, Be-E Aware places greater importance on content narratives that carry significant societal impact, *i.e.*, those that have more interaction with the public (Fig. 15). This prioritization offers users a deeper understanding of conversations that have garnered substantial attention. Moreover, Be-E Aware provides trending keywords, hashtags, and insights into predominant sentiments, empowering healthcare professionals with data-driven insights for making well-informed decisions.

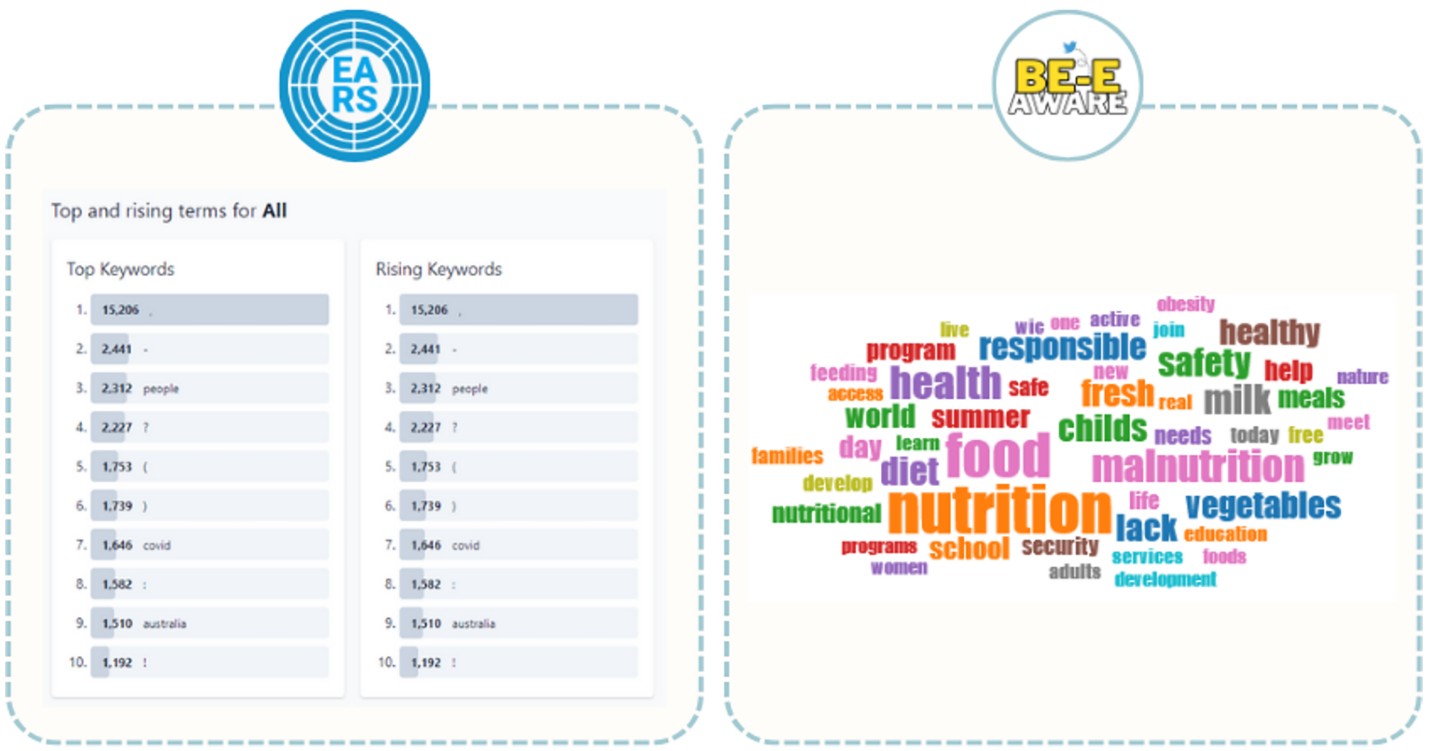

**Figure 14 Comparison of the presentation of top keywords on the EARS and Be-E Aware platforms. Adapted from (*WHO, 2021a*).**

### Advantages and challenges of the OSL system

The Be-E Aware OSL system presents several advantages in combating health misinformation related to NCDs in children. Firstly, it enables real-time monitoring of social media discussions, allowing healthcare professionals to identify trending misinformation early. Secondly, by integrating data from multiple sources, including Twitter and PressReader, the system provides a more comprehensive understanding of online health narratives. Furthermore, the sentiment analysis feature offers insights into public perception, helping tailor more effective health interventions.

However, the system also faces key challenges. Distinguishing between accurate information and misinformation remains complex due to the diverse and often contradictory sources of health content online. A key challenge of the OSL system is distinguishing between legitimate health information and misinformation. Given the diverse sources and varying accuracy of online content, manual verification or AI-powered fact-checking mechanisms could enhance the system's reliability in future iterations.

The collection of social media data, particularly concerning children's health discussions, raises significant privacy and security concerns. While our system follows ethical guidelines by anonymizing data and adhering to Twitter's data usage policies, additional measures, such as data encryption and stricter access controls, could further enhance user privacy.

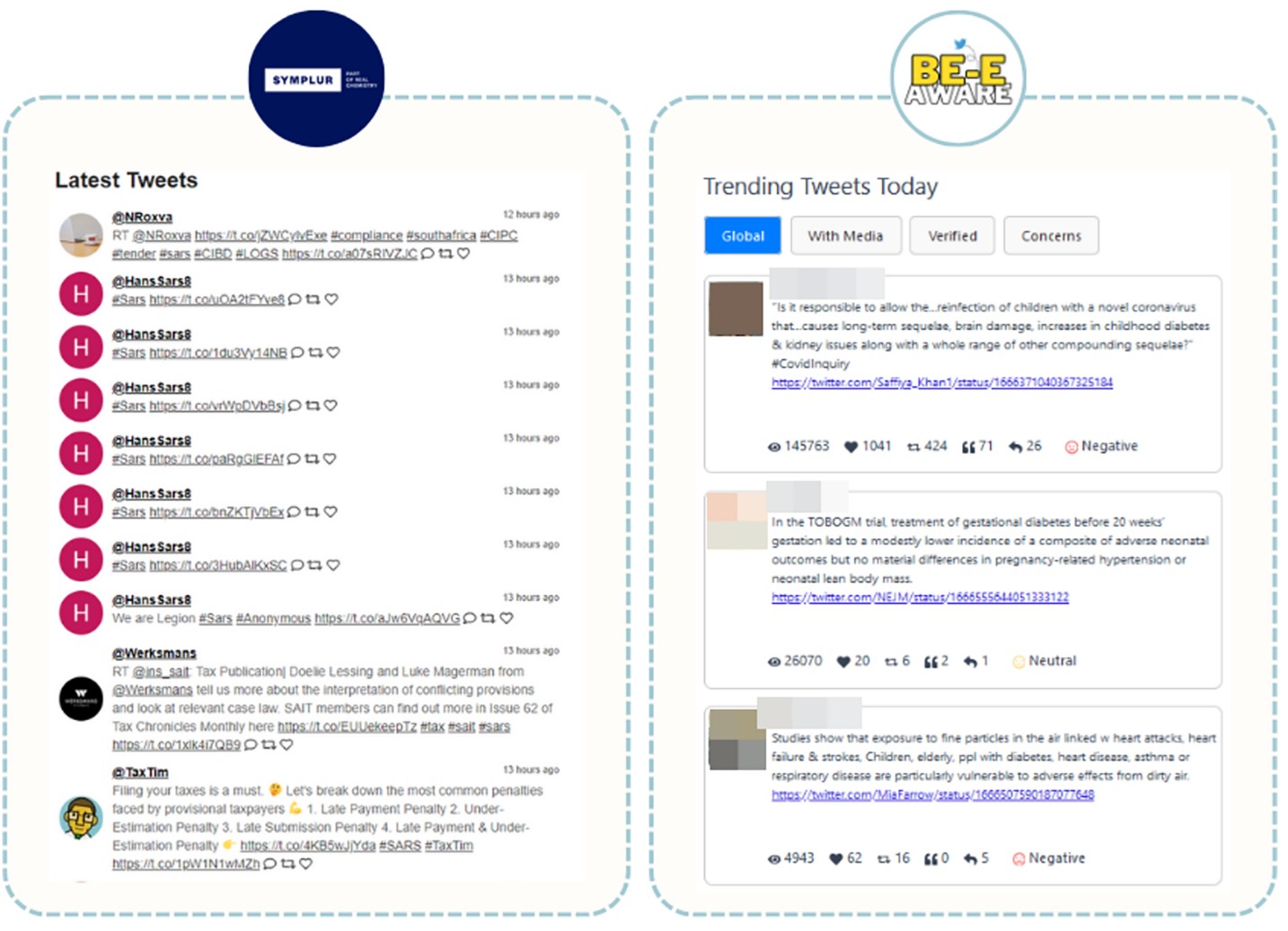

**Figure 15 Comparison of the presentation of Tweets on the SYMPLUR and Be-E Aware platforms. Adapted from** *Symplur (2023)*.

Moreover, reliance on Twitter data limits the scope of the analysis, as other social media platforms may have different patterns of misinformation dissemination. Finally, recent changes to the Twitter API have introduced constraints that could impact the scalability and sustainability of the system in future implementations.

## CONCLUSIONS

The research project collected and processed social media data to potentially contribute to the understanding of NCD, namely in the context of prediction and prevention. The effective implementation of the designed system architecture marked a significant achievement, with extensive data cleaning and processing, ensuring a robust dataset for subsequent analysis. Despite temporary limitations posed by changes in Twitter's API, the project successfully demonstrated the platform's capability to extract valuable information for healthcare professionals and relevant stakeholders.

Beyond healthcare applications, the platform also benefits government authorities by supporting informed decision-making regarding public health policies, resource allocation, and crisis management. Additionally, its health literacy component enables parents and teachers to engage in children's health education, fostering a digitally literate generation equipped with critical thinking skills to discern accurate health information.

While the platform has demonstrated significant potential, certain limitations must be acknowledged. The reliance on external data sources and pre-trained models may affect the accuracy and representativeness of the collected data. Additionally, the system does not operate in real-time, which could impact its responsiveness to emerging health trends. Addressing these limitations in future iterations will be key to enhancing the platform's effectiveness and ensuring its continued relevance in public health surveillance.

This study contributes to the field by introducing an innovative OSL system for tracking NCD-related misinformation in children's health. By integrating multiple data sources and employing sentiment analysis, our approach provides a novel method for healthcare professionals to monitor, assess, and respond to health-related discourse online. This research lays the groundwork for future advancements in AI-driven public health surveillance.

Future work could enhance the platform's capabilities by integrating additional social media sources, expanding language coverage, and implementing location-based filtering mechanisms. The exploration of sentiment analysis in comments and the construction of social network graphs for user interactions could offer a deeper understanding of public sentiment. Advanced features, such as manual tweet classification and machine learning algorithms for automated classification, could further refine the platform. Additionally, providing the platform to a group of diverse agents for familiarization and feedback could offer valuable insights for refining and expanding its usage.

By incorporating PressReader into the OSL system, we can further enhance our understanding of the multifaceted nature of these diseases and identify strategies to address them effectively. The combination of these sources ensures a holistic understanding of the landscape surrounding NCDs in children, capturing both real-time perspectives and validated information from authoritative sources. This integrative approach ensures a more comprehensive analysis of trends, public sentiment, and misinformation patterns, supporting evidence-based strategies to address the challenges posed by these diseases in pediatric populations.

## ACKNOWLEDGEMENTS

All icons used in Figs. 1, 2 and 6 of this manuscript have been designed using resources from Flaticon.com.

### Funding

This work has been supported by FCT (Fundação para a Ciência e Tecnologia) within the Project Scope of the R&D Unity number 00319. This work is intricately interwoven with

the R&D project titled "A Health Promotion Intervention for Vulnerable School Children and Families (BeE-school): A Cluster-Randomized Trial", bearing the reference PTDC/SAU-ENF/2584/2021. There was no additional external funding received for this study. The funders had no role in study design, data collection and analysis, decision to publish, or preparation of the manuscript.

## Grant Disclosures
The following grant information was disclosed by the authors:
FCT (Fundação para a Ciência e Tecnologia): 00319.
R&D Project: PTDC/SAU-ENF/2584/2021.

## Competing Interests
The authors declare that they have no competing interests.

## Author Contributions
- Diana Braga conceived and designed the experiments, performed the experiments, analyzed the data, prepared figures and/or tables, authored or reviewed drafts of the article, and approved the final draft.
- Inês Silva conceived and designed the experiments, performed the experiments, analyzed the data, prepared figures and/or tables, authored or reviewed drafts of the article, and approved the final draft.
- Rafaela Rosário conceived and designed the experiments, performed the experiments, analyzed the data, prepared figures and/or tables, authored or reviewed drafts of the article, and approved the final draft.
- Paulo Novais conceived and designed the experiments, performed the experiments, analyzed the data, prepared figures and/or tables, authored or reviewed drafts of the article, and approved the final draft.
- Hugo Peixoto conceived and designed the experiments, performed the experiments, analyzed the data, prepared figures and/or tables, authored or reviewed drafts of the article, and approved the final draft.
- José Machado conceived and designed the experiments, performed the experiments, analyzed the data, prepared figures and/or tables, authored or reviewed drafts of the article, and approved the final draft.

## Data Availability
Repositories are available at GitHub and Zenodo:

- https://github.com/itsdianabraga/tese (backend)

- Diana, B., Inês, S., Hugo, P., Rafaela, R., Paulo, N., & José, M. (2025). Tweet Analysis Project - OSL. Zenodo. https://doi.org/10.5281/zenodo.15083455

- https://github.com/inessearasilva/bee_school (frontend)

- Inês, S., Diana, B., Hugo, P., Rafaela, R., Paulo, N., & José, M. (2025). bee-school platform. Zenodo. https://doi.org/10.5281/zenodo.15083484

The raw data is available at Zenodo: Hugo, P., Diana, B., Inês, S., Rafaela, R., Paulo, N., & José, M. (2025). Online Social Listening - DataSet [Data set]. Zenodo. https://doi.org/10.5281/zenodo.14900928.

## Supplemental Information

Supplemental information for this article can be found online at http://dx.doi.org/10.7717/peerj.19311#supplemental-information.

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
