# Peer review of "Exploring the potential of online social listening for noncommunicable disease monitoring"

_PeerJ, doi:10.7717/peerj.19311_

## Round 0.1 · original submission · Major Revisions

Based on the reviewers' comments, the manuscript demonstrates relevant potential but requires significant revisions to meet the expected quality for publication. Regarding basic reporting, the manuscript already presents a solid and relevant theoretical foundation, as noted by Reviewer #1. However, Reviewer #2 emphasizes the need to include additional references to further strengthen the study’s theoretical basis. Furthermore, some writing adjustments are necessary, such as dividing a long sentence (lines 113–118) to improve clarity and ensuring proper citation formatting in lines 67, 147, 153, and 159, as highlighted by Reviewer #1.

In terms of the experimental design, both reviewers identified critical gaps. Reviewer #2 suggests the inclusion of standard metrics for evaluating sentiment analysis models, such as ROC and AUC curves, as well as Recall and Precision, which are essential for robust analysis. Reviewer #1 points to missing key information in the introduction, such as data on the prevalence of non-communicable diseases (NCDs) among children and details about children’s use of social media, including age categories, which are essential for establishing the relevance of the study. Additionally, Reviewer #1#3 questions the choice of the OSL platform for disseminating health information and requests clearer justification of the credibility of data sources, such as the Twitter API.

Concerning the validity of findings, the reviewers identified areas requiring greater elaboration. Reviewer #1 highlights that the manuscript does not adequately explain how the strategies for promoting health literacy, which is one of the study's objectives, were developed. There is also a need to clarify the impact of OSL on health literacy, particularly for children aged 6–10. Both reviewers emphasize the importance of specifying the dataset size used for comparing sentiment analysis models, and Reviewer #1 underscores the necessity of including a discussion of the study’s limitations in the conclusion.

Finally, Reviewer #1 and #3 stresses the importance of ensuring that health data presented in the OSL system comes from reliable sources. This reinforces the need for greater methodological transparency and credibility of the data used in the study.

Final Decision: While the manuscript addresses a relevant topic and has potential impact, it requires substantial revisions before it can be reconsidered for publication. The authors should address all the issues raised by the reviewers, paying particular attention to justifying the methods, including additional evaluation metrics, revising the text for greater clarity and completeness, and providing a detailed discussion of the study’s limitations. Resubmission should be accompanied by a detailed response letter outlining the revisions made.

Reviewer 1 ·

Basic reporting

1. The literature used is well-referenced and relevant to the research topic.
2. The sentence in lines 113-118 is too long; it would be better to divide it into two sentences to make it more transparent and easier to understand.
3. The citation in line 67 should use "Li et al. (2020)" following the proper writing format. The same applies to lines 147, 153, and 159.

Experimental design

1. Introduction: From a health perspective, the urgency of the research needs to include the prevalence of NCD issues among children.
2. Introduction: The authors have not explained how many children use online media, including the age categories of children accessing social media. This information is necessary to create relevance for the research topic.
3. State of the Art: It needs to be explained why the OSL platform approach was chosen for health information, considering that children (6-10 years old) also have limited smartphone access. Consider considering my input to state the research question.
4. Method: How credible are the Twitter API news sources that users will read with the expectation of improving health literacy? Please explain.

Validity of the findings

1. One of the objectives of this research is to develop strategies for promoting health literacy; however, how these strategies are formulated has yet to be clearly explained.
2. This research will impact whether the target audience for the information is accurate, such as adolescents. The study aims to provide promotive efforts for children to prevent NCDs. However, please consider the impact of OSL on health literacy for children, particularly those aged 6-10 years.
3. In the conclusion, the authors have yet to explain the limitations encountered in this study.

Additional comments

The author needs to consider credible health data in the OSL system for users to read.

Reviewer 2 ·

Basic reporting

More literature references are required.

Experimental design

For evaluation of various sentiment analysis models ROC and AUC curve should be used.
What is the criteria of choosing the health related keywords?
Why not considered Recall and Precision for evaluation of the models used?

Validity of the findings

What was the data size used in the comparison of sentiment analysis models?

Reviewer 3 ·

Basic reporting

This paper is proposes an Online Social Listening (OSL) system to monitor social media for misinformation on NCDs, focusing on children's health behaviors. By analyzing data from platforms like Twitter and PressReader, it aids healthcare professionals in promoting informed decision-making and instilling healthy habits in children, improving public health outcomes. I think the topic is important and contributive to the promotion of online diagnosis professionals. However, before this paper is published, the following comments should be taken into account when revising the paper.

Experimental design

Major concerns:
1. The authors should provide kernel concept in this study.

2. Pros and cons of proposed OLS should be specified.

3. How to access dataset used in this study? You should briefly introduce the link or access of dataset.

4. While the OSL system can help identify health misinformation, distinguishing between legitimate information and false claims may remain challenging, especially given the diverse sources and varying accuracy levels on social media.

5. Collecting data from social media platforms like Twitter raises concerns about user privacy and the security of sensitive health information, particularly when analyzing children’s health-related discussions.

Validity of the findings

no comment

Additional comments

1. Why no comparison between traditional method and proposed OLS system or other suitable criteria? Please give more illustrations.

2. The contribution of this paper should be highlighted.

---

## Round 0.2 · accepted · Accept

I have carefully assessed the revised version of the manuscript and can confirm that all comments raised by the reviewers have been adequately addressed.
The manuscript is now well-structured, the arguments are clearly presented, and the responses to the initial concerns are thorough and appropriate. I believe that the authors have made a significant effort to improve the paper, and the revised version meets the standards for publication. Therefore, I am pleased to recommend the manuscript for publication in its current form.

Reviewer 1 ·

Basic reporting

The researchers have revised the feedback provided.

Experimental design

It is clear. The researchers included state-of-the-art research and explained their focused approach, making it more understandable.

Validity of the findings

It is clear. In the future, this system will also be able to track references from health journals, further enhancing the accuracy of the information and addressing its limitations.

Reviewer 3 ·

Basic reporting

No further comments.

Experimental design

Sound

Validity of the findings

Sound

Additional comments

None